# Pyrrolizidine Alkaloids from Monofloral and Multifloral Italian Honey

**DOI:** 10.3390/ijerph20075410

**Published:** 2023-04-05

**Authors:** Paola Roncada, Gloria Isani, Mariantonietta Peloso, Thomas Dalmonte, Stefania Bonan, Elisabetta Caprai

**Affiliations:** 1Department of Veterinary Medical Sciences, Alma Mater Studiorum—University of Bologna, via Tolara di sopra 50, 40064 Ozzano dell’Emilia, Italy; 2National Reference Laboratory for Plant Toxins in Food, Food Chemical Department, IZSLER, Via Fiorini, 5, 40127 Bologna, Italy

**Keywords:** honey, Boraginaceae, pyrrolizidine alkaloids, echimidine, health risk assessment

## Abstract

Pyrrolizidine alkaloids (PAs) are secondary metabolites produced by plants as a self-defense against insects. After bioactivation in the liver, some PAs can cause acute or chronic toxicity in humans. The aim of this study was to determine the presence of PAs in 121 samples of monofloral and multifloral honey from three different Italian regions (Friuli-Venezia Giulia, Marche and Calabria) to meet the European Food Safety Authority (EFSA) suggestion. An in-house liquid chromatography with tandem mass spectrometry (LC-MS/MS) method was validated according to European Union Reference Laboratory (EURL) performance criteria. This method allowed the detection and quantification of 35 PAs. Of the 121 honey samples, 38 (31%), mostly from Calabria, contained PAs. The total content of the PAs ranged from 0.9 µg/kg to 33.1 µg/kg. In particular, echimidine was the most prevalent PA. A rapid human exposure assessment to PAs in honey and a risk characterization was performed using the EFSA RACE tool. The assessment highlighted a potential health concern only for toddlers who frequently consume elevated quantities of honey. This study showed a low presence of PAs in Italian honey; however, the importance of continuously monitoring these compounds is stressed, along with the suggestion that the relevant authorities establish maximum limits to guarantee support for producers and consumer safety.

## 1. Introduction

Pyrrolizidine alkaloids (PAs) and their N-oxides (PANOs) are secondary metabolites derived from a necine base produced by plants as a self-defense against insects. Pyrrolizidine alcaloids have received increasing attention due to their toxicity as well as their presence in several plant species relevant to human and animal nutrition [1]. More than 660 different PAs have been identified [2]. They have been detected in more than 6000 plant species [2,3], mainly *Senecio* spp. and *Eupatorium* spp. (Asteraceae), *Echium* spp. (Boraginaceae) and *Crotalaria* spp. (Fabaceae) [4]. The PA content can vary depending on the plant species, site of accumulation, harvest time and climatic conditions. In general, they are found in greater quantities in flowers and seeds and to a lesser extent in leaves, stamens, and roots. The possible routes of human dietary exposure to PAs occur through the ingestion of plants and herbal products (drugs, herbal teas, dietary supplements) [5,6,7,8] as well as animal products such as honey [4]. Honey and other hive products can be contaminated with PAs as a results of bees foraging on alkaloid-producing plants [9]. Recent studies have shown that PAs can also be found in water and soil [10,11].

The toxicity of PAs has been widely documented, being almost exclusively associated with their metabolites [1]. Pyrrolizidine alkaloids themselves are pro-toxins, biologically and toxicologically inactive, and need to be metabolically activated in order to exert toxicity; consequently, if not activated, they do not develop toxicity [5]. These compounds are mainly bioactivated in the liver by CYP450 monooxygenases [1,2,4,12] producing 6,7-dihydro-7-hydroxy-1-(hydroxymethyl)-5H-pyrrolizine (ester pyrrolyl), a strong electrophile that can rapidly bind to nucleophilic centers such as nucleic acids, proteins and amino acids, forming pyrrole complexes that can persist in tissues and generate toxicity, especially in the liver [2,9,13,14,15]. These compounds have been shown to be hepatotoxic, pneumotoxic, genotoxic and carcinogenic and exhibit developmental toxicity [5,7]. An important detoxification pathway is by conjugation with glutathione, forming soluble compounds that are much less toxic and more easily eliminated [16].

Pyrrolizidine alkaloids are hepatotoxic to animals and humans; they can cause acute toxicity and have chronic effects [9,12,17]. Chronic exposure to low levels of PAs can cause liver cirrhosis and cancer as metabolic activation produces genotoxic and carcinogenic reactive pyrrolic forms [9]. The International Agency for Research on Cancer (IARC) has evaluated several PAs and has classified lasiocarpine, monocrotaline and riddelliine as Group 2B (possibly carcinogenic to humans), while hydroxysenkirkine, isatidine, jacobine, retrorsine, seneciphylline, senkirkine and symphytine were included in Group 3 (not classifiable as to their carcinogenicity to humans) [15,18,19]. As a consequence, the EFSA, the European Food Safety Authority, has repeatedly considered the issue of the presence of pyrrolizidine alkaloids in food with the aim of establishing the level of risk to public health [12,13].

In fact, maximum levels in bee pollen have been set by the Commission Regulation (EU) 2020/2040 [1,20], while there is still no regulation for the presence of these alkaloids in honey and limits to establish criteria for acceptance or rejection in the marketing of this food have not yet been set [21]. For this reason, the EFSA has suggested collecting data regarding the content of PAs/PANOs in honey of different geographical and botanical origins. Exhaustive data regarding PA/PANO content in honey from Italy are scarce. Lucatello et al. (2021) recently found 17 PAs/PANOs in honey samples from the Veneto region and showed that 45% of the samples analyzed contained at least one PA. However, the consumption of this honey did not seem to represent a risk for adult consumers [22]. 

To meet the EFSA suggestion, the aim of this research was to determine the presence of PAs in Italian honey. Honey samples from three Italian regions (Friuli-Venezia Giulia, Marche, and Calabria) were analyzed for their PA/PANO content using an in-house liquid chromatography with tandem mass spectrometry (LC-MS/MS) method, developed and validated by the National Reference Laboratory for Plant Toxins in Food (LNR-TVN) of the Istituto Zooprofilattico Sperimentale della Lombardia e dell’Emilia Romagna (IZSLER) in Bologna. This method is capable of detecting up to 35 analytes, according to Commission Regulation (EU) 2020/2040 [20]. The assessment of human exposure to PAs in honey has been estimated on the basis of the results of the analysis in order to characterize the health risk for all age groups of the population.

## 2. Materials and Methods

### 2.1. Sampling

A total of 121 different types of honey were collected from several Italian beekeepers in three Italian regions: Friuli-Venezia Giulia (*n* = 37), Marche (*n* = 39) and Calabria (*n* = 45). The province of origin of the honey is reported in Table 1. Samples included both multifloral (*n* = 34) and monofloral (*n* = 87) honey. Monofloral honey was the most representative and the plant species are reported in Table 2. 

### 2.2. Chemicals and Standards

Analytical standards of all pyrrolizidine alkaloids and their N-oxide were purchased from Phytolab (Vestenbergsgreuth, Germany): Echimidine (Em), Echimidine-N-oxide (EmNO), Echinatine (En), Echinatine-N-oxide (EnNO), Erucifoline-N-oxide (ErNO), Europine (Eu), Europine-N-oxide (EuNO), Heliosupine (Hs), Heliosupine-N-oxide (HsNO), Heliotrine (Ht), Heliotrine-N-oxide (HtNO), Indicine (Id), Indicine-N-oxide (IdNO), Integerrimine (Ir), Integerrimine-N-oxide (IrNO), Intermedine (Im), Intermedine-N-oxide (ImNO), Lasiocarpine (Lc), Lycopsamine (Ly), Lycopsamine-N-oxide (LyNO), Retrorsine (Rt), Retrorsine-N-oxide (RtNO), Rinderine (Rn), Rinderine-N-oxide (RnNO), Senecionine (Sn), Senecionine-N-oxide (SnNO), Seneciphylline (Sp), Seneciphylline-N-oxide (SpNO), Senecivernine (Sv), Senecivernine-N-oxide (SvNO), Senkirkine (Sk), Spartioidine (St), Spartioidine-N-oxide (StNO), Usaramine (Us) and Usaramine-N-oxide (UsNO). Methanol (LC-MS grade) was from VWR Chemicals (Rosny-sous-Bois-cedex, France), sulphuric acid (96%) and acetonitrile (LC-MS grade) were from Carlo Erba Reagents (Val de Reuil Cedex, France), formic acid was from Carlo Erba Reagents (Milan, Italy) and ammonium formate (analytical grade) was from Sigma-Aldrich (St. Louis, MO, USA). Ultrapure water used throughout the experiments was produced by an EvoQua Water Technologies system (Diessechem, Milan, Italy).

### 2.3. Materials

QuEChERS reagents (magnesium sulphate 4 g, sodium chloride 1 g, sodium citrate 1 g, disodium hydrogen citrate sesquihydrate 0.5 g) were from Agilent Technologies (Santa Clara, CA, USA).

### 2.4. Working Solutions

Stock standard solutions of each compound were prepared by dissolving suitable quantities of reference material in methanol to obtain a concentration of 1000 µg/mL. Working standard solutions containing all PAs—native and N-oxide—were prepared in water/methanol (95:5 v/v) for spiking purposes. All solutions were stored at −20 °C [23]. 

### 2.5. Sample Preparation

A 2.5 ± 0.1 g aliquot of homogenized sample was weighed into a 50 mL Falcon tube.

Samples were treated as follows: addition of 15 mL of sulphuric acid 0.1 M, vortex and horizontal shaker for 45 min; addition of 15 mL of acetonitrile and QuEChERS extraction reagents, horizontal shaker for 30 min and centrifuged at 3000 rpm at room temperature for 10 min. An aliquot of supernatant extract was dried with a gentle flow of nitrogen in a water bath at 40 °C. The dry extract was dissolved in 1 mL of water/methanol (95:5 v/v) and transferred into vial for LC-MS/MS analysis. A quality control sample (i.e., spiked sample at LOQ) was assessed at every batch analysis.

### 2.6. Melissopalynological Analysis

To identify the botanical and geographical origin of honey, a qualitative-quantitative melissopalynological analysis was carried out using the microscopic method UNI 11299:2008. The analysis was performed on four honey samples showing higher quantities of PAs: two multifloral (27.1–33.1 µg/kg) and one chestnut (30.6 µg/kg) from the Calabria and one Stachys honey (9.2 µg/kg) from the Marche. All honey samples were diluted in ultrapure water and centrifuged. The sediment was transferred onto the microscope slide to be examined. For the estimation of the relative frequencies of pollen types, a minimum of 300 pollen grains were counted at 400x magnification using light microscope Axiolab 5 (Carl Zeiss, Jena, Germany) [24].

### 2.7. Instrumentation 

The LC-MS/MS system used was an Acquity ultra-performance liquid chromatograph (UPLC) coupled to Quattro Premiere XE triple quadrupole mass spectrometer (Waters, Milford, MA, USA). The system was computer-controlled and data acquisition, peak integration and calibration were performed using TargetLynx software v.4.1. The chromatographic column was an Acquity UPLC C8 100 cm × 2.1 mm, 1.7 µm (Water Corporation, Milford, CT, USA). The mobile phase consisted of 5 mM ammonium formate and 0.1% formic acid in water (A) and in methanol (B) [25]. The flow rate was set at 0.3 mL/min; 10 μL was injection volume. The mobile phase gradient was set as follows: from 5% to 20% of B for 10 min, from 20% to 50% for 5 min and return to initial condition for 0.5 min and hold for 1.5 min. Total run time was 17 min. The ESI source was in positive ionization mode with a capillary voltage of 1.0 kV, a cone voltage of 40 V, a source temperature of 120 °C and a desolvation temperature of 450 °C. Ionization and fragmentation conditions for PAs were identified by using a continuous infusion of the tuning solutions and gradually changing the parameters. 

### 2.8. Quantification 

Pyrrolizidine alkaloids were identified and quantified on the basis of retention time, ion fragments produced and ion ratio. The retention time had to be ± 0.2 min compared to reference peaks. In Figure 1 and Figure 2 are shown the chromatograms of a standard mixture of PAs and a standard mixture of PANOs, respectively, at 5 ng/mL concentration. 

A honey matrix-matched calibration curve was prepared to quantify PA content of unknown samples. Pyrrolizidine alkaloid concentration was extrapolated by means of the least squares regression method. Calibration curve concentration levels were 0.5–1–2.5–5–10–25 ng/mL. 

### 2.9. Performance Evaluation

The method was validated according to the EURL-MP guidance document plant toxins performance criteria [26] and Regulation 401/2006/EC [27]. Specificity, recovery rates, linearity, repeatability, within-laboratory reproducibility, limit of detection (LOD) and limit of quantification (LOQ) were evaluated. Specificity was verified and the presence of interference was checked by analyzing 20 honey samples of different species. Correlation coefficients (R^2^) of the matrix-matched calibration curve had to be ≥ 0.99 for all analytes. The LOD at 0.5 µg/kg and LOQ at 1 µg/kg were established on the basis of a signal-to-noise ratio, S/N = 3 (LOD) and S/N ≥ 5 (LOQ). Repeatability and overall recovery were assessed by analyzing blank samples fortified with PAs at concentrations of 1–10–25 µg/kg in six replicates per level. The same experiment was carried out in two additional sessions to determine within-laboratory reproducibility. The selectivity of the LC–MS/MS method is obtained by acquiring the data in MRM mode and monitoring one precursor ion and two daughter ions for each molecule according to SANTE/12089/2016 [28]. The LC-MS/MS parameters are reported in Table 3. Of the 35 analytes, 14 were isomers and were classified into five groups: Sn-group (Sn, Sv, Ir), Ly-group (Ly, Im, Id, En, Rn), Sp-group (Sp, St), Em-group (Em, Hs) and Rt-group (Rt, Us); the same applied to the N-oxides. 

### 2.10. Statistical Analysis

The lower-bound approach was used to substitute with zero the values below the LOQ, as indicated by EFSA [12]. The central limit theorem was taken into consideration to assess the normality of distribution [29]. Kruskal–Wallis one-way ANOVA with Dunn’s multiple comparisons post-hoc test was performed to determine significant differences among the honey types in the same region of origin and among the honey types without taking into consideration the origin. The *p*-value obtained underwent Bonferroni’s correction. The Levene test was performed to assess the homoscedasticity and consequently the robust one-way ANOVA Tahmane post-hoc test was applied to detect significant differences of the PA/PANO content among the three different regions. 

The presence/absence of PAs was used to create a dichotomous variable and X^2^ test was performed to detect significant association between the above-mentioned variable, the honey types and the region of origin. The type and the strength of association were assessed by calculating the Cramer V value [30]. Statistical analyses were performed using R 4.2.1 (R foundation for statistical computing; Vienna, Austria; https://www.R-project.org/ accessed on 15 January 2023). Data are reported as median, mean ± SD (standard deviation). A *p*-value < 0.05 was considered statistically significant.

### 2.11. Exposure Assessment and Risk Characterization

Human exposure assessment and risk characterization were performed using the EFSA RACE tool [31]. It is a spreadsheet that calculates human intake of food contaminants (e.g., PAs) for all member state subgroups of population taking into account: (1) food consumption data [32]; (2) detailed food description based on FoodEx2 food classification; (3) experimental occurrence of selected substances in the food commodity. In order to evaluate the health risk, exposure is therefore compared with the relevant toxicological reference points. The first is the Acute Reference Dose (ARfD), which is an estimate of the quantity of a substance in food and/or drinking-water, normally expressed on a body-weight basis, that can be ingested in a period of 24 h or less without appreciable health risk to the consumer on the basis of all the known facts at the time of the evaluation [33]. The Acute Reference Dose is a reference point for short-term exposure. The second is the benchmark dose lower confidence limit for a 10% excess cancer risk (BMDL_10_), a reference value for long-term exposure that corresponds to a specific change in an adverse response compared to the response in unexposed subjects [34]. 

For contaminants such as PAs, these two toxicological values for both acute and chronic exposure were considered. The acute human health risk was characterized in terms of percentage of ARfD ingested while chronic risk was described with the Margin of Exposure (MoE) approach. The MoE is the ratio between the dose associated with a small increase in adverse effect (BMDL_10_) and the level of human exposure calculated by RACE.

For RACE exposure assessment, the selected food item was “honey” while the highest PA content detected in the samples was entered as our worst-case occurrence data. The ARfD was 2 mg/kg bw per day and BMDL_10_ was 237 µg/kg bw, as updated reference points for the sum of 1,2-unsaturated PAs assuming equal potency [13].

A percentage of ARfD ingested lower than 100% and an MoE of 10,000 or higher would be of low concern from a public health point of view for acute and chronic risk, respectively. The RACE tool analyzed every population subgroup in terms of consumption pattern. This approach results in outcomes referencing the whole population or consumers only depending on the answers given within the food consumption survey. For all population groups, mean, median and 95th percentile output values were calculated. The 95th percentile was considered the relevant value for a high honey consumption pattern.

## 3. Results and Discussion

### 3.1. LC-MS/MS Method Validation

Pyrrolizidine alkaloids (*n* = 18) and their related PANOs (*n* = 17) were detected and quantified. In the validation phase, the possible coelution of alkaloid isomers was assessed and the following co-elutions were found according to Commission Regulation (EU) 2020/2040: lycopsamine/indicine, renderine/echinatine, intermedine-N-oxide/indicine-N-oxide, senecivernine/integerrimine, echimidine/heliosupine, seneciphylline/spartioidine, seneciphylline-N-oxide/spartioidine-N-oxide, retrosine/usaramine and retrosine-N-oxide/usaramine-N-oxide. In chromatograms of blank honey extracts no significant interfering peaks were detected at the retention time of all 35 PAs/PANOs. The method exhibited linearity for PA concentrations in the 1 to 50 µg/kg range; R squared (R^2^) was ≥ 0.99 for all the PAs in honey. In accordance with the EURL-MP guidance document for plant toxin performance criteria [35], a recovery range of 70–120%, a relative standard deviation (RSD_r_ %) of repeatability ≤ 20% and an RSD_R_% of within laboratory reproducibility ≤ 25% were required. The experimental values of all the substances are reported in Table 4 and all the performance criteria were met. Therefore, the method can be considered fit for purpose. The limit of quantification (LOQ) for all PAs and PANOs was 1 µg/kg. 

### 3.2. PA/PANO Content in Honey Samples

The method was used to determine 35 PAs in 121 honey samples from three different Italian regions. The samples were analyzed in six analytical sessions and a quality control (QC) sample was evaluated simultaneously, having a recovery in the 70–120% range. The percentage of PA-positive samples is reported in Figure 3. Figure 4 shows the compounds detected in the honey samples and their sum when more than one PA was present while, in Figure 5, the mean content of the PAs is reported.

Of the 37 honey samples analyzed from Friuli-Venezia Giulia, PAs were found in concentrations above the LOQ in only 4 multifloral samples (11%). The predominant PAs found in three samples (8%) were echimidine (mean ± SD 4.94 ± 3.47 µg/kg) followed by seneciphylline in one sample (3%, 12.7 µg/kg) while in the Marche only one sample of monofloral honey from *Stachys officinalis* contained echimidine above the LOQ (9.2 µg/kg) (Figure 6). *S. officinalis* can be considered a typical crop of the Marche. It belongs to the family of Lamiaceae and does not belong to one of the major PA-producing families; therefore, the presence of an echimidine content above the LOQ was probably due to pollen contamination by the PA-producing plant species, as confirmed by melissopalynological analysis. Thirty-three samples from Calabria contained PAs above the LOQ. In particular, all 9 multifloral samples were contaminated, while 24 samples of the 36 monofloral honey samples were contaminated. The predominant PAs were echimidine (31 samples) followed by lycopsamine (5 samples), intermedine (3 samples), echinatine N-oxide (2 samples), rinderine N-oxide (1 sample) and heliosupine N-oxide (1 sample) (Figure 4).

Overall, the total content of PAs ranged from 0.9 µg/kg in a chestnut honey from Calabria to 33.1 µg/kg in a multifloral sample from the same region (Figure 4). The mean content (±SD) of the PAs detected was 7.11 ± 8.25 µg/kg, of the same order of magnitude, albeit higher, than the mean value of 4.7 ± 11.1 µg/kg reported by Lucatello et al. (2021) for honey samples from local producers in the Veneto [22]. 

Of the 35 PAs analyzed, 8 showed values higher than the LOQ (Figure 5). The most abundant and most variable PA was echimidine, detected in 35 samples (27%) with a mean content (±SD) of 6.28 ± 6.76 µg/kg, followed by lycopsamine, detected in 5 samples (4%), intermedine in 3 samples (2.5%), echinatine N-oxide in 2 samples (1.7%) and finally seneciphylline, echimidine N-oxide, rinderine N-oxide and heliosupine N-oxide detected in 1 sample each (0.8%). One sample from Calabria contained four different PAs: echimidine, lycopsamine, intermedine and echinatine N-oxide. In particular, echimidine, lycopsamine and intermedine were the most prevalent PAs. This finding was in agreement with previous publications [36,37,38,39,40,41,42]. Table 5 summarizes the PA/PANO data present in the scientific literature.

Pollen composition was investigated in order to understand the origin of PAs in the most contaminated Calabria honey and in the *Stachys officinalis* honey sample from the Marche. Melissopalynological analysis revealed the presence of pollen from the genus belonging to the Boraginaceae family: *Echium, Cerinthe* and *Cynoglossum*. Echimidine is a typical alkaloid of the genus *Echium*, in particular of *Echium plantagineum*, a plant widespread throughout Italy (Portal to the Flora of Italy, https://dryades.units.it/floritaly/index.php?procedure=taxon_page&tipo=all&id=4297 (accessed on 7 February 2023)). The elevated levels of echimidine and its N-oxide derivative, which were higher than the LOQ in the honey from all three regions, could be explained by the notable presence of this species in Italy. Lycopsamine is another alkaloid that was present in the samples, although in lower quantities. This alkaloid is produced by a number of plants, including *Echium vulgaris*, belonging to the Boraginaceae family and widespread in Italy (Portal to the Flora of Italy, https://dryades.units.it/floritaly/index.php?procedure=taxon_page&tipo=all&id=4297 (accessed on 7 February 2023)).

The results obtained in the present study regarding Italian honey were consistent with the results published by other authors who have reported the concentration of PAs in European honey. A study of 40 samples of Polish multifloral honey collected directly from beekeepers showed an alkaloid content ranging from 1.0 to 20.2 μg/kg, with an average content of 2.9 μg/kg. In the same study, the analysis of 14 honey samples of Asian origin showed a much higher content of PAs [43]. In a more recent study [44], the PA content in the Polish honey ranged from 2.2 to 31.6 μg/kg while the total PA content monitored in foreign honey ranged from 5.8 to 147.0 μg/kg.

Bodi et al. (2014) published the results of PA content in honey sampled from German and Austrian beekeepers. These samples showed a significantly lower rate of positive samples than those bought at the supermarket and from other sources. The mean total PA content ranged from 6.1 μg/kg of honey in beekeeper samples to 14 μg/kg in discount products and 15 μg/kg in branded honey [45]. Dübecke et al. (2011) observed substantial differences in the quantity of PAs found in honey depending on the country of origin. The mean concentration of PAs in 381 European honey samples was 17 μg/kg, including negative samples [37]. These data were very similar to the present data. Honey from Germany, Bulgaria and Romania showed lower levels of PAs (1–43 μg/kg) than honey from Italy and Spain (concentrations up to 225 μg/kg) as honey from these regions often contained an elevated number of *Echium* pollen grains (18 PAs and N-oxides monitored) [46].

Martinello et al. (2014) reported a higher content of PAs (nine monitored alkaloids) in honey samples that were blends of EU and non-EU honey for which the mean content determined was 17.5 μg/kg. The mean PA content in the EU honey was 3.1 μg/kg [41]. All the results reported, however, could have been underestimated due to the limited number of PAs monitored.

Honey samples produced in Belgium were less contaminated and presented a different contamination profile; PAs were found in 67% of the samples examined, with maximum and average concentrations of 60.53 μg/kg and 1.20 μg/kg, respectively. The majority of samples (49%) contained from 0.05 to 0.99 μg/kg of contaminants [47]. 

The analysis of 103 Spanish honey samples (*Echium* spp. honey) showed the presence of PAs in 97 samples with a content ranging from 1 to 237 μg/kg. The mean PA concentration of the PA-positive samples was 48 ± 55.5 μg/kg. The PA pattern was clearly dominated by echimidine, lycopsamine and their N-oxides, which accounted for 97.8% of the total ΣPA, while seneciphyilline and heliothrine N-oxide were detected at a much lower incidence [42].

### 3.3. Variables Affecting the Content of PAs/PANOs in Honey Samples

Qualitative variables: a significant, albeit moderate (Cramér’s V = 0.483), association (*p* = 1.56 × 10^−13^) was detected between the presence/absence of PAs and the specific region using the Chi-square test. In Calabria, 73% of the samples were found to contain PAs. 

Quantitative variables: significant differences among honey types were detected within the same region. In the Marche, the sum of the PAs and echimidine were significantly higher in Stachys honey (*p* = 7.53 × 10^−06^), while in honey samples from Friuli-Venezia Giulia no significant differences were recorded. A more complex pattern was evidenced in honey from Calabria due to the presence of a high percentage of samples having a PA content > the LOQ; this made multiple comparisons among the different types of honey possible. In particular, the multifloral honey samples showed a significantly higher content of echimidine (10.28 ± 10.69 µg/kg) than citrus fruit honey (1.7 ± 1.25 µg/kg) (*p* = 0.0261) (Figure 7) and a significantly higher content of intermedine than the acacia honey (*p* = 0.0253). Finally, one monofloral Sulla honey from the province of Catanzaro was characterized by the exclusive presence of N-oxide derivatives of three PAs, echinatine, rinderine and heliosupine, which were not present in any other honey sample (Figure 4).

### 3.4. Exposure Assessment and Health Risk Characterization

Data entries on the RACE spreadsheet are summarized in Table 6 while the outcomes of the calculation are reported in Table 7 and Table 8 for acute and chronic exposure to PAs, respectively.

For acute exposure assessment, all values were well below the ARfD. On the other hand, for chronic assessment the RACE tool calculated a MoE lower than 10,000 (i.e., mean 7302, median 7876) for toddler consumers. The number of observations (< 60) was not sufficient for an accurate MoE calculation for the 95th percentile output for toddlers, both as consumers and as total population. However, these values were below the 10,000 threshold (3759 and 4654, respectively). This means that the daily consumption of elevated quantities of honey containing PAs at the highest concentration detected could probably pose a health risk for toddlers and children. Unfortunately, the RACE tool does not give specific information regarding food consumption; however, looking at the Italian food consumption survey, which is the RACE reference for Italy, it can be seen that means of 16.6 g/day for the total population and 19.6 g/day for consumers are data referring to the category “sugar, fructose, honey and other nutritious sweeteners”. For consumers of large quantities, these values can double [32]. Furthermore, this survey is quite outdated as it was carried out in 2005–2006 and consumption habits have changed over time. 

The health concerns for toddlers and children, who are frequent consumers of large quantities of honey, were also highlighted by the 2011 EFSA opinion regarding PAs [12]; however, it should be noted that a previous BMDL_10_ of 70 µg/kg bw was used for the risk characterization while the mean occurrence in honey could be considered in line with the highest findings of the authors. The exposure assessment and risk characterization were carried out according to the worst-case scenario, having used a single occurrence datum of the PAs detected in a single sample. In order to better understand whether there is a real risk associated with the consumption of honey, additional PA monitoring should be carried out. The margins of exposure for all other population groups were greater than 10,000, signifying a negligible risk for those age groups.

## 4. Conclusions

This study, along with others carried out in different countries, could be very useful for both Food Safety Authorities and beekeepers in identifying, classifying and creating a map of the distribution of geographical areas at risk for the presence of PA-producing flora. In addition, health authorities need to develop better traceability of the origin of honey, together with the integration of data on nomadism practices, so that geographic areas at risk for the presence of PA-containing plants may easily be identified. This identification could be very useful in improving the safety and quality of honey.

While a sum of 33.1 µg PAs/kg has been associated with negligible health risks related to the consumption of honey, chronic exposure assessment and risk characterization have highlighted a potential health concern only for toddlers who frequently consume elevated quantities of honey. This finding could be influenced by uncertainties deriving from real honey consumption habits and a “worst-case” PA occurrence. In order to better characterize the risk, additional monitoring studies regarding PAs should be implemented. It is worth pointing out that the ingestion of honey could be associated with infant botulism as honey is a dietary reservoir of *Clostridium botulinum* spores [48]. This possibility is well known to pediatricians who should not recommend honey-containing supplements or the use of honey as a flavoring agent for infants, in particular those younger than 12 months [48,49]. 

Despite the possible health risks for specific population subgroups, honey is a very rich food, possessing health and therapeutic properties that vary, similarly to its aroma, depending on the flowers from which the bees have extracted the nectar. In addition to glucose and fructose, honey contains polyphenols, flavonoids, alkaloids, glycosides and volatile compounds with proven antioxidant, anti-microbial and anti-inflammatory effects and with potential neuroprotective effects [50,51,52,53].

## Figures and Tables

**Figure 1 ijerph-20-05410-f001:**
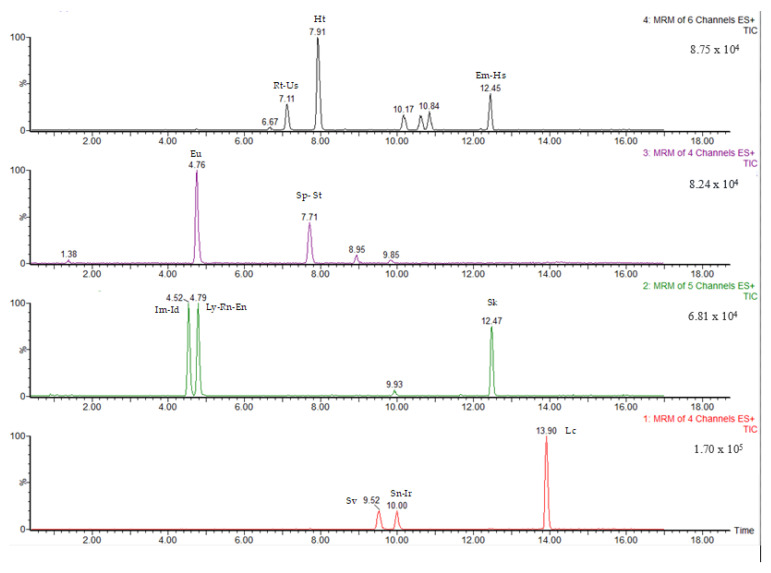
Chromatogram of a standard mixture of PAs at 5 ng/mL concentration.

**Figure 2 ijerph-20-05410-f002:**
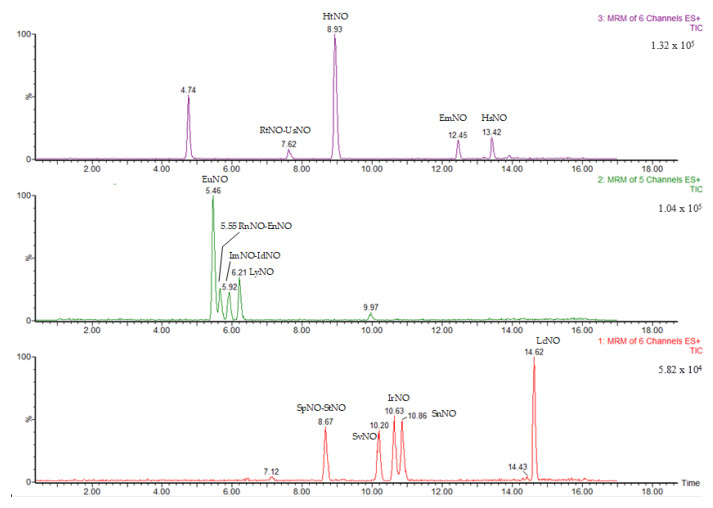
Chromatogram of a standard mixture of PANOs at 5 ng/mL concentration.

**Figure 3 ijerph-20-05410-f003:**
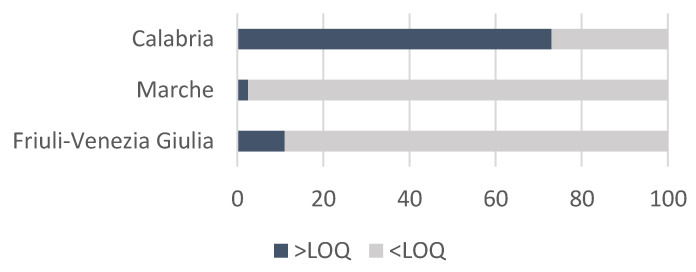
The percentage of honey samples with PA content >LOQ and <LOQ is reported to highlight the differences among regions.

**Figure 4 ijerph-20-05410-f004:**
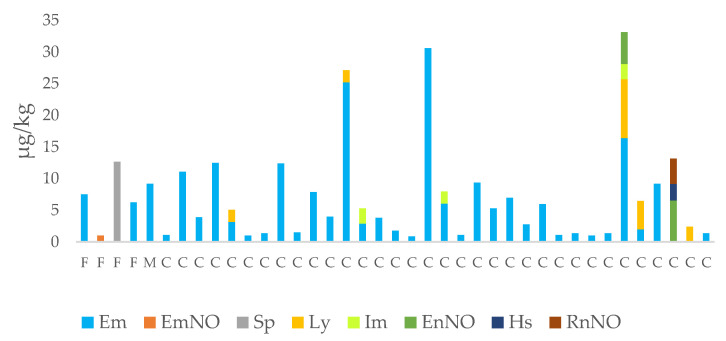
Pyrrolizidine alkaloids (PAs) in honey samples from different Italian regions (F = Friuli-Venezia Giulia; M = Marche; C = Calabria). Data expressed in µg/kg, Em = echimidine; EmNO = echimidine N-oxide; Sp = seneciphylline; Ly = lycopsamine; Im = intermedine; EnNO = echinatine N-oxide; Hs = heliosupine; RnNO = rinderine N-oxide.

**Figure 5 ijerph-20-05410-f005:**
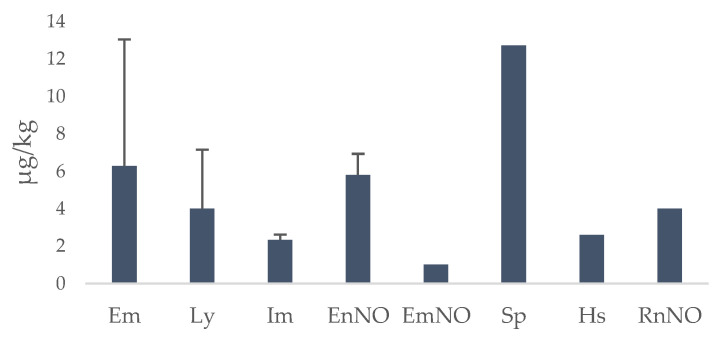
Mean content of pyrrolizidine alkaloids (PAs) in honey samples from Friuli-Venezia Giulia, Marche and Calabria Italian regions. Data are expressed in µg/kg and reported as mean ± SD, with the exception of EmNO, Sp, Hs and RnNO, which were found in only one sample. Em = echimidine (*n* = 35); EmNO = echimidine N-oxide (*n* = 1); Sp = seneciphylline (*n* = 1); Ly = lycopsamine (*n* = 5); Im = intermedine (*n* = 3); EnNO = echinatine *n*-oxide (*n* = 2); Hs = heliosupine (*n* = 1); RnNO = rinderine *n*-oxide (*n* = 1).

**Figure 6 ijerph-20-05410-f006:**
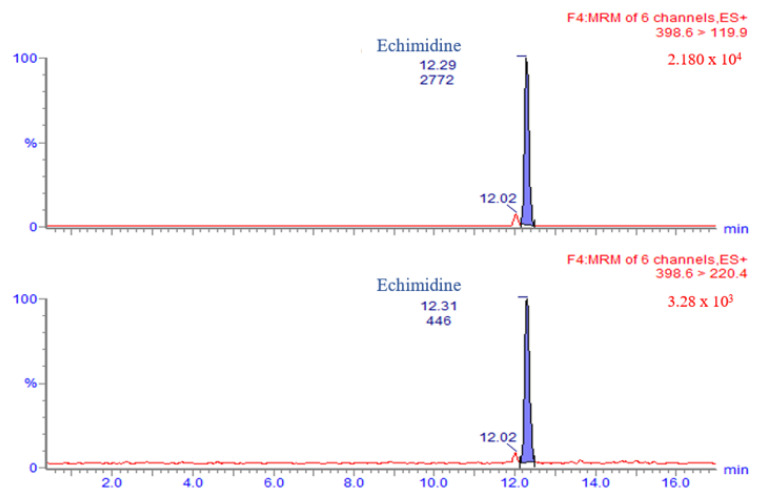
Chromatograms of echimidine (9.2 µg/kg) present in a *Stachys officinalis* honey sample.

**Figure 7 ijerph-20-05410-f007:**
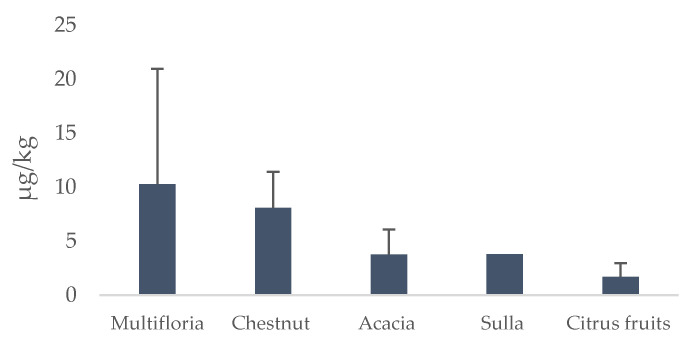
Mean content of echimidine in samples from Calabria region. Data are expressed in µg/kg and reported as mean ± SD.

**Table 1 ijerph-20-05410-t001:** Botanical characteristics and origin of honey samples.

Honey Types	Origin
Multifloral, Winter heath, Common whitebeam, Dandelion, Acacia, Chestnut and Linden	Udine (Friuli-Venezia Giulia)
Multifloral, Rapeseed, Linden, Acacia, Bastard Indigobush and Chestnut	Pordenone (Friuli-Venezia Giulia)
Multifloral and Mahaleb Cherry	Trieste (Friuli-Venezia Giulia)
Multifloral, Sunflower, Rapeseed, Honeydew, Mustard, Clover and Betony	Ancona (Marche)
Multifloral, Chestnut and Acacia	Fermo (Marche)
Multifloral and Sunflower	Macerata (Marche)
Multifloral, Chestnut, Honeydew and Acacia	Ascoli Piceno (Marche)
Multifloral, Acacia, Sulla, Citrus Fruits and Chestnut	Vibo Valentia (Calabria)
Acacia, Citrus Fruits and Chestnut	Cosenza (Calabria)
Multifloral, Acacia, Sulla, Citrus Fruits, Eucalyptus and Chestnut	Catanzaro (Calabria)
Chestnut and Winter heath	Crotone (Calabria)
Acacia and Citrus Fruits	Reggio Calabria (Calabria)

**Table 2 ijerph-20-05410-t002:** Botanical origin of monofloral honey samples.

Monofloral Honey Types	Total Samples
Acacia or robinia (*Robinia pseudoacacia*)	25
Chestnut (*Castanea sativa*)	15
Sunflower (*Heliantus annuus L*.)	14
Citrus fruits	7
Linden (*Tilia L*.)	6
Rapeseed (*Brassica napus*)	4
Honeydew	3
Winter heath (*Erica carnea*)	2
Sulla (*Sulla coronaria*)	2
Eucalyptus *(E. camaldulensis, E. occidentalis)*	2
Mustard (*Sinapis arvensis*)	1
Clover (*Trifolium pratense*)	1
Betony (*Stachys officinalis*)	1
Common whitebeam (*Sorbus aria*)	1
Bastard indigobush (*Amorpha fruticosa*)	1
Dandelion (*Taraxacum officinale*)	1
Mahaleb cherry (*Prunus mahaleb*)	1

**Table 3 ijerph-20-05410-t003:** The LC-MS/MS parameters for PAs and PANOs (CE: collision energy, Q: Quantifier ion, q: qualifier ion).

Pyrrolizidine Alkaloids	MH^+^	CE	m/z	Q, q
Sn-group	336.2	25 25	120.2 138.0	Q q
Ly-group	299.7	20 25	138.0 156.0	Q q
Ht	314.1	20 25	138.0 156.0	Q q
Eu	330	20 15	138.0 156.0	Q q
Sk	366.1	30 25	122.0 167.9	Q q
ErN	366.1	35 25	94.1 119.1	Q q
LyN-group	316.1	25 25	172.0 138.0	Q q
HtN	330.2	25 25	172.0 111.0	Q q
EuN	346.2	25 20	172.0 328.1	Q q
Lc	412.1	25 18	120.1 220.0	Q q
Sp-group	334	25 25	120.1 138.0	Q q
Em-group	398.6	20 15	119.9 220.4	Q q
Rt-group	352.1	25 25	120.0 138.3	Q q
SnN-group	352.1	25 30	94.0 118.0	Q q
LcN	428.1	30 25	254.0 94.0	Q q
SpN-group	350.1	30 25	94.0 120.0	Q q
EmN-group	414.2	30 25	254.0 220.0	Q q
RtN-group	368.3	30 20	94.0 120.0	Q q

**Table 4 ijerph-20-05410-t004:** Validation parameters for all 35 PAs and PANOs (RSD_r_: repeatability, RSD_R_: within-laboratory reproducibility).

Analyte	Conc (µg/kg)	Recovery %	RSDr	RSD_R_
Em-Hs	1	94.5	13.53	14.81
	10	85.3	13.21	23.08
	25	76.9	4.52	6.28
EmN	1	82.9	10.79	11.63
	10	75.4	10.04	10.75
	25	72.9	9.41	11.21
Eu	1	77.5	14.83	17.97
	10	60.3	4.47	18.61
	25	84.8	0.96	23.87
EuN	1	83.6	13.24	15.28
	10	79.3	4.18	16.96
	25	71.9	5.33	10.58
Ht	1	90.6	11.06	12.52
	10	75.7	8.91	14.42
	25	70.4	5.91	7.13
HtN	1	91.3	11.45	12.87
	10	80.6	4.48	5.16
	25	71.8	8.19	9.81
Im	1	79.9	13.17	20.19
	10	70.1	2.95	19.46
	25	62.6	5.28	7.50
ImN	1	74.3	0.69	15.41
	10	71.1	7.12	9.81
	25	67.1	17.75	9.66
Lc	1	93.7	7.01	15.83
	10	84.3	11.65	23.03
	25	76.2	4.81	7.64
LcN	1	74.8	17.86	20.46
	10	68.3	12.96	13.77
	25	67.6	4.51	9.87
Ly, Id, En, Rn	1	84.9	7.72	20.84
	10	72.5	6.81	24.21
	25	65.6	3.64	13.70
LyN, IdN	1	74.8	9.56	9.99
	10	68.3	5.54	8.86
	25	67.6	7.52	11.24
Rt, Us	1	75.8	11.34	16.64
	10	71.5	7.89	19.31
	25	64.7	7.28	7.63
RtN, UsN	1	87.6	8.92	21.43
	10	74.7	8.36	10.56
	25	66.7	6.95	11.48
Sp, St	1	80.3	15.31	19.67
	10	87.6	7.01	23.97
	25	79.6	6	9.28
SpN, StN	1	79.7	11.9	19.64
	10	70.2	6.51	14.11
	25	65.2	6.41	11.89
Sn, Ir	1	96.4	10.37	11.41
	10	84.9	6.45	18.22
	25	81	5.18	5.87
SnN	1	85.1	11.37	13.61
	10	78.5	5.72	12.22
	25	74.2	2.88	12.05
Sv	1	88.1	12.1	22.88
	10	71.9	8.39	20.51
	25	72	4.6	20.16
SvN	1	97.9	9.38	10.21
	10	85.8	8.14	11.68
	25	84.5	7.83	10.81
Sk	1	87.4	12.85	18.87
	10	76	11.11	17.55
	25	77.5	2.58	16.36
EnN	1	92.1	12.85	20.56
	10	75.3	11.11	9.22
	25	61.9	2.58	10.03
RnN	1	76.1	14.45	15.31
	10	70.5	5.86	17.63
	25	68.9	8.37	13.12
IrN	1	88.4	6.77	12.48
	10	81.6	4.28	7.22
	25	77.2	10.13	13.36
HsN	1	95.1	8.64	10.21
	10	86.4	7.7	9.04
	25	87.7	3.49	11.66

**Table 5 ijerph-20-05410-t005:** Content of PAs/PANOs detected in honey in different studies. Maximum levels or, when possible, range or average values have been reported. The PAs/PANOs covered by this work are listed (n.i.: not investigated).

	Kowalczyk et al., 2022		Picron et al., 2020		Martinello et al., 2017	Lucatello et al., 2016	Griffin et al., 2014	Martinello et al., 2014	Orantes-Bermejo et al., 2013		This Research
PA/PANO (µg/kg)	Polish Honey	Foreign Honey	Belgian Honey	Foreign Honey	Retail Honey	Italian Beekeepers	Retail Honey	Retail Honey	Spanish Honey by Beekeepers		Italian Honey
Echimidine	7.3	120.0	5.91	8.84	0.4–3.3	0.3–1.0	545.5	169	36.9 ± 44.36	237	1.0–30.6
Echinatine	n.i.	n.i.	n.i.	n.i.	n.i.	n.i.	n.i.	n.i.	n.i.	n.i.	<LOQ
Europine	n.i.	n.i.	0.009	134.85	n.i.	n.i.	n.i.	n.i.	n.i.	n.i.	<LOQ
Heliosupine	n.i.	n.i.	n.i.	n.i.	n.i.	n.i.	n.i.	n.i.	n.i.	n.i.	LOQ-2.6
Heliotrine	n.i.	n.i.	<LOQ	39.44	<LOQ	<LOQ	<LOQ	<LOQ	n.i.	n.i.	<LOQ
Indicine	n.i.	n.i.	n.i.	n.i.	n.i.	n.i.	n.i.	n.i.	n.i.	n.i.	<LOQ
Integerrimine	n.i.	n.i.	n.i.	n.i.	n.i.	n.i.	n.i.	n.i.	n.i.	n.i.	<LOQ
Intermedine	9.2	23.3	n.i.	n.i.	<LOQ	<LOQ	n.i.	31	n.i.	n.i.	1.9–2.4
Lasiocarpine	n.i.	n.i.	<LOQ	5.77	<LOQ	<LOQ	n.i.	<LOQ	n.i.	n.i.	<LOQ
Lycopsamine	14.1	22.5	n.i.	n.i.	0.2–74.7	<LOQ	392.6	42	5.7 ± 4.28	18	1.9–9.3
Retrorsine	4.3	4.3	5.82	5.32	<LOQ	0.9–14.5	<LOQ	<LOQ	n.i.	n.i.	<LOQ
Rinderine	n.i.	n.i.	n.i.	n.i.	n.i.	n.i.	n.i.	n.i.	n.i.	n.i.	<LOQ
Senecionine	2.2	2.7	9.67	1.46	<LOQ	0.8–2.1	8.4	<LOQ	n.i.	n.i.	<LOQ
Seneciphylline	4.1	4.0	7.35	4.04	<LOQ	0.6–1.1	5.7	<LOQ	4.1 ± 4.79	20	LOQ-12.71
Senecivernine	<LOQ	3.0	4.15	0.41	n.i.	n.i.	n.i.	n.i.	n.i.	n.i.	<LOQ
Senkirkine	n.i.	n.i.	42.44	1.15	<LOQ	<LOQ	<LOQ	<LOQ	<LOQ	<LOQ	<LOQ
Spartioidine	n.i.	n.i.	n.i.	n.i.	n.i.	n.i.	n.i.	n.i.	n.i.	n.i.	<LOQ
Usaramine	n.i.	n.i.	n.i.	n.i.	n.i.	n.i.	n.i.	n.i.	n.i.	n.i.	<LOQ
Echimidine-N-oxide	n.i.	n.i.	8.24	0.17	n.i.	n.i.	n.i.	n.i.	21.4 ± 23.09	70	LOQ-1.01
Echinatine-N-oxide	n.i.	n.i.	n.i.	n.i.	n.i.	n.i.	n.i.	n.i.	n.i.	n.i.	5–6.6
Erucifoline-N-oxide	n.i.	n.i.	0.09	0.14	n.i.	n.i.	n.i.	n.i.	n.i.	n.i.	<LOQ
Europine-N-oxide	n.i.	n.i.	0.54	1.30	n.i.	n.i.	n.i.	n.i.	n.i.	n.i.	<LOQ
Heliosupine-N-oxide	n.i.	n.i.	n.i.	n.i.	n.i.	n.i.	n.i.	n.i.	n.i.	n.i.	<LOQ
Heliotrine-N-oxide	n.i.	n.i.	<LOQ	0.37	n.i.	n.i.	n.i.	n.i.	2.3 ± 0.58	4	<LOQ
Indicine-N-oxide	n.i.	n.i.	0.47	0.18	n.i.	n.i.	n.i.	n.i.	n.i.	n.i.	<LOQ
Intermedine-N-oxide	n.i.	n.i.	0.26	0.21	n.i.	n.i.	n.i.	n.i.	n.i.	n.i.	<LOQ
Integerrimine-N-oxide	n.i.	n.i.	n.i.	n.i.	n.i.	n.i.	n.i.	n.i.	n.i.	n.i.	<LOQ
Lycopsamine-N-oxide	n.i.	n.i.	0.29	0.09	n.i.	n.i.	n.i.	n.i.	4.0 ± 2.79	8	<LOQ
Retrorsine-N-oxide	n.i.	n.i.	0.86	0.90	n.i.	n.i.	<LOQ	n.i.	<LOQ	<LOQ	<LOQ
Rinderine-N-oxide	n.i.	n.i.	n.i.	n.i.	n.i.	n.i.	n.i.	n.i.	n.i.	n.i.	LOQ-4
Senecionine-N-oxide	n.i.	n.i.	0.54	0.26	n.i.	n.i.	<LOQ	n.i.	<LOQ	<LOQ	<LOQ
Seneciphylline-N-oxide	n.i.	n.i.	0.16	0.28	n.i.	n.i.	<LOQ	n.i.	<LOQ	<LOQ	<LOQ
Senecivernine-N-oxide	n.i.	n.i.	0.67	<LOQ	n.i.	n.i.	n.i.	n.i.	n.i.	n.i.	<LOQ
Spartioidine-N-oxide	n.i.	n.i.	n.i.	n.i.	n.i.	n.i.	n.i.	n.i.	n.i.	n.i.	<LOQ
Usaramine-N-oxide	n.i.	n.i.	n.i.	n.i.	n.i.	n.i.	n.i.	n.i.	n.i.	n.i.	<LOQ

**Table 6 ijerph-20-05410-t006:** Summary of RACE parameters for exposure assessment and risk characterization.

RACE Parameters	
Contaminant	Pyrrolizidine alkaloids (sum of 1,2-unsaturated)
Food description	Honey
Analytical result	33.1 µg/kg
Reference value acute (ARfD)	2 mg/kg bw
Reference value chronic (BMDL_10_)	237 µg/kg bw
Survey country	Italy [32]

**Table 7 ijerph-20-05410-t007:** Summary of RACE outputs for acute health risk characterization.

Output (% ARfD)			
Acute Consumers Only	Mean	Median	95th Percentile
Toddlers	0.0022	0.0024	0.0047 ^1^
Other children	0.0012	0.0008	0.0024 ^1^
Adolescents	0.0007	0.0005	0.0016 ^1^
Adults	0.0005	0.0003	0.0011
Elderly	0.0004	0.0003	0.0010
Very elderly	0.0006	0.0005	0.0009 ^1^
Acute whole population	Mean	Median	95th percentile
Toddlers	0.00023	- ^2^	0.00236
Other children	0.00008	- ^2^	0.00072
Adolescents	0.00001	- ^2^	-
Adults	0.00003	- ^2^	0.00018
Elderly	0.00003	- ^2^	0.00026
Very elderly	0.00004	- ^2^	0.00031

^1^ Number of observations lower than 60; the 95th percentile may not be statistically robust. ^2^ Values not given by the model.

**Table 8 ijerph-20-05410-t008:** Summary of RACE outputs for chronic health risk characterization.

Output (MoE)			
Chronic Consumers Only	Mean	Median	95th Percentile
Toddlers	7303	7876	3759 ^1^
Other children	12,496	15,752	5783 ^1^
Adolescents	24,134	24,702	17,184 ^1^
Adults	36,740	50,476	12,411
Elderly	44,500	42,961	20,764 ^1^
Very elderly	27,294	31,147	7279 ^1^
Chronic whole population	Mean	Median	95th percentile
Toddlers	52,581	- ^2^	4654
Other children	141,868	- ^2^	16,468
Adolescents	851,578	- ^2^	- ^2^
Adults	456,875	- ^2^	62,293
Elderly	391,059	- ^2^	39,953
Very elderly	327,532	- ^2^	37,591

^1^ Number of observations lower than 60; the 95th percentile may not be statistically robust. ^2^ Values not given by the model.

## Data Availability

Not applicable.

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
