# Peer review of "Pyrrolizidine Alkaloids from Monofloral and Multifloral Italian Honey"

_ijerph, 2023, doi:10.3390/ijerph20075410_

Round 1
Reviewer 1 Report
The present study entitled "Pyrrolizidine Alkaloids from Monofloral and Multifloral Italian Honey" is a work that exposes a necessary topic and whose presence in the bibliography is scarce.
The PA content in honey is reflected by its botanical origin and this has to be studied and related to it. For this reason, its study is of great value to typify a product such as honey and carry out a good labeling for the consumer.
However, honey also contains numerous benefits that show that its consumption is beneficial for human health. In fact, not all the honeys studied in this work contained PAs, (probably the honeys without Echium contribution) so I think the somewhat alarmist approach of the work should be softened.
On the other hand, I believe a greater description of the pollen content (palynological analysis) is necessary and relate it to these compounds. would improve understanding of the results.
I have included in the Pdf that I have attached some suggestions for the authors to take into account. I consider that the work can be published in the journal after making this minor revision.

Author Response
We would like to thank the reviewer for the time spent in the revision. Our answers are reported below.
The PA content in honey is reflected by its botanical origin and this has to be studied and related to it. For this reason, its study is of great value to typify a product such as honey and carry out a good labeling for the consumer. However, honey also contains numerous benefits that show that its consumption is beneficial for human health. In fact, not all the honeys studied in this work contained PAs, (probably the honeys without Echium contribution) so I think the somewhat alarmist approach of the work should be softened.
R: Thank you for the suggestion. We have modified sentences with alarmist approach.
On the other hand, I believe a greater description of the pollen content (palynological analysis) is necessary and relate it to these compounds. would improve understanding of the results.
R: The pollen was analysed in few samples, those with high PA content. A sentence was added. We will accept your suggestion for the next studies and carry out the melissopalynological analysis to all honey samples.
I have included in the Pdf that I have attached some suggestions for the authors to take into account. I consider that the work can be published in the journal after making this minor revision.
R: thank you for your suggestions which have been all satisfied.

Reviewer 2 Report
This is a generally readable paper on a topic of interest. It is fairly well presented but requires clarification in places, especially in terms of the statistical methods and some of those results. The English requires improving in numerous places, many of which are marked on the manuscript along with other suggestions and queries. One reference needs editing. These required edits are generally minor.

Author Response
We would like to thank the reviewer for the time spent in the revision. Our answers are reported below.
This is a generally readable paper on a topic of interest. It is fairly well presented but requires clarification in places, especially in terms of the statistical methods and some of those results. The English requires improving in numerous places, many of which are marked on the manuscript along with other suggestions and queries. One reference needs editing. These required edits are generally minor.
R: The authors thanks the reviewer for the suggestions all of which were met.
In particular, English has been checked and corrected using a certified service.
Section 2.9"Statistical analysis" has been modified.
The sentences on lines 35-352 and on 405-410 have been edited and made clearer.
The word "percentile" has been added and the reference #25 has been corrected.

Reviewer 3 Report
March 14, 2023
Environmental Research and Public Health
Dear Editors,
Thank you for the opportunity to review “Pyrrolizidine Alkaloids from Monofloral and Multifloral Italian Honey.” This paper is novel work reporting the DHPA content of honey from three Italian production zones. These results were then compared with reports from other countries and continents. It is well written and will be an excellent contribution to your Journal. Would suggest a couple changes to simplify and make these results more available the reader.
Specific Suggestions:
45- Nucleic acids might be better than DNA as RNA is also affected.
133+ Hopefully you have a chemist looking at these sections.
272, 277, 283- these figures are difficult to interpret. There is little information to give the reader an understanding of the variability. The data from Figures 3 and 4 information could easily be in a table with means and standard deviation so that readers could appreciate the variation. Figure 5 has error bars, but there is no explanation to those results that don’t have bars.
322+ These comparisons are important and a table with the results from these different studies would facilitate side by side comparison.
201- I would be helpful if there were better explanation of the EFSA Race tool and what the results mean. This might be best in the discussion as some work is needed to explain its biases comparing exposures that have intermittent chronic effects.
Author Response
We would like to thank the reviewer for his appreciation of our research. Our answers are reported below.
45- Nucleic acids might be better than DNA as RNA is also affected.
R: the reviewer is right and we accepted the suggestion
133+ Hopefully you have a chemist looking at these sections.
R: certainly. the analyses were performed at the National Reference Laboratory for Plant Toxins in Food (LNR-TVN) of the IZSLER in Bologna.
201- I would be helpful if there were better explanation of the EFSA Race tool and what the results mean. This might be best in the discussion as some work is needed to explain its biases comparing exposures that have intermittent chronic effects.
R: an explanation has been added
272, 277, 283- these figures are difficult to interpret. There is little information to give the reader an understanding of the variability. The data from Figures 3 and 4 information could easily be in a table with means and standard deviation so that readers could appreciate the variation.
R: we have deleted figure 3a, which was less informative. As for the data reported in figure 3b and 4, we believe that graphical representation is more immediate than a table, especially for figure 4 which shows which PAs were present in each positive sample. Moreover, these are individual data and it is not possible calculate mean and SD.
Figure 5 has error bars, but there is no explanation to those results that don’t have bars.
R: the reviewer is right. The lack of the error bar in the last data is due to the fact that these are individual data. The explanation was added to the caption of the figure.
322+ These comparisons are important and a table with the results from these different studies would facilitate side by side comparison.
R: A table was added following the suggestion of the reviewer.

Reviewer 4 Report
Manuscript title:
Pyrrolizidine Alkaloids from Monofloral and Multifloral Italian Honey
This study has certain significance in food science and human exposure……... However, revisions are necessary for the current version of the manuscript. The following questions to be addressed/considered may be helpful to improve the manuscript. Major comments • In the abstract, it would be even better to have a sentence as a future perspective. • The unit/abbreviation is not mentioned before, consider defining the abbreviation when mentioned for the first time…. Please check throughout the manuscript to define the abbreviations. • Lake of scientific literature to support the statements and findings throughout the manuscript…... I have made some suggestions for that and more need it…. • More information is needed for ALL TABLE captions and define the abbreviation and units that are used. And adjust the significant figures for the table and manuscript. • Grammar and punctuation issuers need to be addressed. I have selected/mentioned some as examples. • I have a major concern about the results and discussion section. The authors describe the results and compare the results with previous studies, however, insight mechanisms are still insufficient. Specific comments: Abstract Line 12-13: where in insects? Or human? Please state! If the unit/abbreviation is not mentioned before, consider defining the abbreviation when mentioned for the first time. Introduction: Line 26: PAs are derived from necine base – please correct it Line 35: Most recently there are reports that exposure through water is possible, I suggest to add that. Line 38: this is the repetition of lines 25-26. Please revise or delete. Line 51-54: A complicated sentence, please revise and check the grammar Line 58-61: These are rather long sentences, better to break them down into more sentences. Line 70-71: Reference is needed here. In MM section Literature references are missing for all sub-section. It would be better to cite the references that the procedure adopted. Additional info is needed for the table caption, most importantly significant figures. In MM section, what is the quality control (QC) data? There is no mention of the QC. What is the accuracy of the instruments, recovery, LOD, and LOQ ……. These parameters are needed to report the efficiency of any analytical system. In general, how many times you’ve recorded the data,? duplicate? Triplicate?..... what you mentioned in the text is not clear, please elaborate more on this Table 3: why ‘’ PYRROLIZIDINE ALKALOIDS’’ are in capital – if there is no critical reason they all should be small letters R&D section These sections are repeating information already presented and explain things in an unnecessarily complicated way. The quality of the manuscript would benefit from the whole section being condensed, Line 253-270, Line 322-346, Line 377-404… Figure 3. Nice figure! Do we have error bars? It is worth adding the error bars…. Line 377-390: Reference is needed here. Conclusion Important conclusions!
Author Response
We would like to thank the reviewer for the time spent in the revision. Our answers are reported below.
Major comments •
In the abstract, it would be even better to have a sentence as a future perspective.
R: done
The unit/abbreviation is not mentioned before, consider defining the abbreviation when mentioned for the first time…. Please check throughout the manuscript to define the abbreviations.
R: done
Lake of scientific literature to support the statements and findings throughout the manuscript…... I have made some suggestions for that and more need it
R: following the suggestion of the reviewer we added references
More information is needed for ALL TABLE captions and define the abbreviation and units that are used. And adjust the significant figures for the table and manuscript.
R: done
Grammar and punctuation issuers need to be addressed. I have selected/mentioned some as examples.
R: English has been checked and corrected using a certified service.
I have a major concern about the results and discussion section. The authors describe the results and compare the results with previous studies, however, insight mechanisms are still insufficient.
R: the authors do not understand what insight mechanisms the reviewer is referring to. In any case, the purpose of the research is to evaluate the presence of PAs in Italian honey.
Specific comments:
Abstract Line 12-13: where in insects? Or human?
R: human was added
If the unit/abbreviation is not mentioned before, consider defining the abbreviation when mentioned for the first time.
R: done
Introduction:
Line 26: PAs are derived from necine base – please correct it
R: done
Line 35: Most recently there are reports that exposure through water is possible, I suggest to add that.
R: we have added a sentence and two references on the presence of PA in water and soil
Line 38: this is the repetition of lines 25-26. Please revise or delete.
R: the sentence has been deleted.
Line 51-54: A complicated sentence, please revise and check the grammar. Line 58-61: These are rather long sentences, better to break them down into more sentences.
R: English has been checked and corrected using a certified service
Line 70-71: Reference is needed here.
R: the reference was already present; anyway, the sentence has been made clearer.
In MM section Literature references are missing for all sub-section. It would be better to cite the references that the procedure adopted.
R: the requestes references were added
Additional info is needed for the table caption, most importantly significant figures.
R: additional info was added to table and figure captions
In MM section,
what is the quality control (QC) data? There is no mention of the QC. What is the accuracy of the instruments, recovery, LOD, and LOQ ……. These parameters are needed to report the efficiency of any analytical system. In general, how many times you’ve recorded the data,? duplicate? Triplicate?..... what you mentioned in the text is not clear, please elaborate more on this Table 3: why ‘’ PYRROLIZIDINE ALKALOIDS’’ are in capital – if there is no critical reason they all should be small letters
R: all the suggestions/concerns have been met.
R&D section
These sections are repeating information already presented and explain things in an unnecessarily complicated way. The quality of the manuscript would benefit from the whole section being condensed.
R: the sentences were condensed and simplified
Line 253-270, Line 322-346, Line 377-404… Figure 3. Nice figure! Do we have error bars? It is worth adding the error bars….
R: we have deleted figure 3a, which was less informative. In figures 3 and 4, mean and standard deviation cannot be reported because individual data are reported. The lack of the error bar in the last data of FIg. 5 is due to the fact that these are individual data. The explanation was added to the caption of the figure.
Line 377-390: Reference is needed here.
R: We are reporting our results; we do not understand what references is needed
Conclusion
Important conclusions!
R: The authors would thank the reviewer for the appreciation

Round 2
Reviewer 4 Report
The revised manuscript has improved compared to the original version. The authors tried to address my questions as much as possible. I recommend the manuscript to be published!